# Integrated Effect of Plastic Mulches and Biorational Insecticides in Managing Tomato Chlorotic Spot Virus (TCSV) and Its Vector Thrips in Tomatoes

**DOI:** 10.3390/insects14090740

**Published:** 2023-09-03

**Authors:** Rafia A. Khan, Dakshina R. Seal, Shouan Zhang, Oscar E. Liburd, James Colee

**Affiliations:** 1Tropical Research and Education Center, University of Florida-IFAS, Homestead, FL 33031, USA; dseal3@ufl.edu (D.R.S.); szhang0007@ufl.edu (S.Z.); 2Entomology and Nematology Department, University of Florida, Steinmetz Hall, 1881 Natural Area Dr., Gainesville, FL 32611, USA; oeliburd@ufl.edu; 3Consultant, Statistical Consulting Unit, Institute of Food and Agricultural Sciences, University of Florida, Gainesville, FL 32611, USA; colee@ufl.edu

**Keywords:** thrips, tomato chlorotic spot virus, plastic mulch, biorational insecticides, management

## Abstract

**Simple Summary:**

Thrips transmitted tomato chlorotic spot virus (TCSV) is one of the limiting factors of tomato production in South Florida. After invading in 2012, growers are mostly dependent on conventional and broad-spectrum insecticides to face this challenge. Insecticide alone is inadequate to manage this pest-borne disease. A combined effect of cultural and chemical control strategy can be a potential management approach for this pest without imposing any environmental stress. The present study was conducted to determine the effect of four different plastic mulches and four biorational insecticides on the abundance of thrips population, marketable yield, and presence of TCSV-infected tomatoes. Reflective plastic mulches, especially silver on black (S/B) and Entrust^®^SC, among the biorational insecticides used, were the most effective in managing the thrips population while increasing the marketable yield and reducing the TCSV-infected tomatoes.

**Abstract:**

In the USA, tomato chlorotic spot virus (TCSV) was first identified in Miami-Dade County of Florida in 2012. This viral disease is transmitted by thrips (Thysanoptera: Thripidae) of different species, imposing a serious threat to the entire tomato production in the state. Both cultural and chemical control techniques could be essential tools to combat this vector-borne disease. In the present two-year-long study, we determined the effect of different types of plastic mulches and biorational insecticides on managing thrips and TCSV. Results from the leaf and flower samples showed a significantly lower adult thrips population in Entrust^®^SC treated tomatoes than in other treated and untreated tomatoes in 2018. Silver on black and silver on white reflective plastic mulches significantly reduced the adult thrips population in 2018. In both study years, marketable yield was significantly higher in tomatoes treated with Entrust^®^SC and reflective plastic mulches than in other treatments. The incidence of TCSV was significantly reduced in tomatoes treated with Entrust^®^SC and reflective plastic mulches than the untreated control in 2018. Marketable yield was negatively correlated with the thrips population, as observed from the Pearson correlation coefficient analysis. This research describes a potentially viable management program for thrips and thrips-transmitted TCSV.

## 1. Introduction

Tomato chlorotic spot virus (TCSV) is an emerging tospovirus in the USA, first identified in South Florida in 2012 [1]. The virus has caused considerable yield loss in tomatoes since its invasion in South Florida. Tomato plants can be infected with TCSV at three weeks after transplantation. The affected plant shows the symptoms of necrotic lesions and chlorotic spots followed by terminal stem and leaf death, wilting, necrosis, and deformation of leaves [2,3]. More TCSV-infected plants are observed at the edge of the tomato fields and near the external inoculation source, such as weedy areas, fallow lands, nurseries, and other crop production areas [4,5]. The prevalence of the spread of TCSV has been reported mostly from South and Central Florida. However, TCSV has been identified elsewhere in the US, including Ohio [6] and New York [7], indicating the probability of TCSV spreading beyond Florida. Tomato chlorotic spot virus is primarily vectored by western flower thrips (WFT) (*Frankliniella occidentalis* Pergande) and common blossom thrips (CBT) (*F. schultzei* Trybom). Both species have been reported from the vegetable production area of South Florida and Puerto Rico with the incidence of TCSV [8,9,10,11]. Thrips, especially WFT, CBT, melon thrips (*Thrips palmi* Karny), and Florida flower thrips (*Frankliniella bispinosa* Morgan) are commonly found in the field crops of South Florida, where TCSV is more prevalent.

An integrated disease management approach should be applied to manage both tospovirus and their vector thrips. Crucial strategies include the use of resistant cultivars, effective insecticides to suppress the vector thrips, and cultural practices interrupting the thrips-virus–plant interactions. TSW-resistant cultivars have the potential to minimize the loss from TCSV. Although these resistant tomato cultivars for TCSV were identified, their field performance in South Florida, including fruit yield, quality, and tolerance to other important diseases (such as bacterial spot), has not been determined [12]. Insecticides are essential tools to address thrips control and virus transmission. Repeated application of insecticides may cause the development of resistance in pests [13], the resurgence of pests, the development of secondary pests [14], elimination of natural enemies [15], biomagnification [16], and impact on nontarget species [17]. Moreover, the effectiveness of insecticides is variable when they are applied for vector control in different agroecosystems. For an effective insecticide-oriented management strategy, it is vital to understand the transmission process and the interaction between insects and their agro-chemical environment [18]. Insecticides aldicarb (Temik^®^) and phorate (Thimet^®^) were promising in suppressing thrips population and decreasing thrips feeding injury, thus reducing spotted wilt incidence in peanut [19,20]. The organophosphate insecticide Thimet^®^ is thought to induce a host defense response against virus replication [21]. However, high mammalian toxicity renders these two insecticides less desirable for their application in agriculture [22]. Recently, the production of Temik^®^ was ceased by the agreement of the United States Environmental Protection Agency (US-EPA) and Bayer CropScience (Leverkusen, Germany) [23]. Spinosad, a biological insecticide derived from an actinomycete bacterium with a unique mode of action (IRAC group 5) and low mammalian toxicity, appeared to be an effective insecticide for managing thrips [24,25]. A recent study reported reduced spinosad activity in managing thrips due to the frequent use of this product to manage multiple pests on various vegetable crops [26]. Application of acibenzolar-S-methyl (Actigard^®^) induces systemic acquired resistance against pathogens. Actigard^®^ was found useful in tomato spotted wilt virus (TSWV) reduction in tobacco and tomato [27,28]. Scientists [29] found that the neonicotinoid insecticides acetamiprid and thiamethoxam, along with spinosad, were effective in managing WFT in sweet pepper, tomato, and lettuce. Spinosad, fipronil, and methamidophos were found to be effective against thrips adults and larvae, whereas spirotetramat was found to affect only thrips larvae [30].

The use of metalized and ultra-violet (UV) reflective plastic mulches can also be an effective way to manage thrip and tospovirus prevalence and increase the yield per unit area [31,32,33,34,35,36]. Moreover, the use of plastic mulch to cover the beds is an effective tool for migrating thrips population only at the early developmental stage of plants with small canopy [35,37]. At the late growth stage of tomato with a dense canopy, the reflection capacity of plastic mulch becomes greatly limited. Metalized plastic mulch showed a lower incidence of TSWV in tomatoes than the plants grown on black plastic mulch [38,39]. A different study on silver and other colored plastic mulches on disease incidence indicates their effectiveness in thrips reduction, lowering or delaying the occurrence of tospovirus, and increased yield [28,40,41,42,43,44]. The objective of the present study was to determine the effectiveness of different polyethylene mulches (‘silver on black’, ‘silver on white’, ‘black on black’, ‘white on black’) and biorational insecticides such as Aza-direct^®^ (azadirachtin), Entrust^®^ SC (Spinosad), Grandevo (*Chromobacterium subtsugae* strain PRAA4-1^T^), and Spear^®^T (GS-omega/kappa-Hxtx-Hv1a) on thrips abundance, TCSV incidence, marketable yield and the volume of fruit production of tomato in South Florida. 

## 2. Materials and Methods

### 2.1. Time and Location of the Study, Plant Material, and Field Preparation

We conducted all field trials on the research farm (25°30′33.7″ N 80°30′17.1″ W) at the Tropical Research and Education Center (TREC), UF/IFAS, Homestead, FL, from November to March 2018 and repeated the study from December to April 2019. This field study was conducted using tomato (*Solanum lycopersicum* cv. Sanibel) as the main crop. The ‘Sanibel’ tomato transplants were donated by Mobley Plant World, LLC, Labelle, FL, USA. The soil type of the field was Krome gravelly loam (Loamy-skeletal, carbonatic hyperthermic lithic Udorthents), consisting of about 67% limestone pebbles (>2 mm) and 33% finer particles [45]. The field was prepared by standard commercial practices using a moldboard plow (CASE International, Felton, DE, USA) and disking (Athens Disc Machine, Athens, TN, USA). The raised beds, each 0.91 m 0.91 m (3-foot) wide and 0.15 m (6-inch) high with 1.82 m (6-foot) spacing between center to center of two adjacent beds were prepared by a machine (Kennco Manufacturing Inc., Ruskin, FL, USA). Before covering the beds with plastic mulch, a granular fertilizer (N-P-K:8-16-16) (TomatoGain 8-16-16 Tomato Plant Food, Bougainvillea Growers International, St. James City, FL, USA) was applied at 1500 kg/hectare in a furrow 20 cm from and parallel to both sides of the transplant row at the center of the bed and incorporated within 15 cm of the soil surface. Halosulfuron methyl (0.5 oz/acre, Sandea^®^, Group#2, Gowan Company LLC., Yuma, AZ, USA) was used as a pre-emergence herbicide to control weeds. Irrigation was provided through two drip tapes (Ro-Drip, St. Joseph, MI, USA) with 30 cm emitter spacing placed 15 cm apart on each side parallel to the center of a bed. Experimental plots were then covered with different plastic mulches. Tomato seedlings were transplanted 45 cm apart at the center (transplant row) of each bed and 1.82 m between beds 21 d after the application of halosulfuron methyl. Research plots were 7.62 m (25 feet) long and 1.82 m (6 feet) wide, with a 1.52 m (5 feet) buffer between treatment plots with 15 plants. 

### 2.2. Plastic Mulches, Insecticide Treatments, and Experimental Design

Different plastic mulches used in this study included ‘silver on black’ (S/B), ‘silver on white’ (S/W), ‘black on black’ (B/B), and ‘white on black’ (W/B) (Can-Grow XSB, 0.9 mils, Canslit, Inc., Victoriaville, QC, Canada, and supplied by Imaflex, Inc., Thomasville, NC, USA). We used no mulch (bare soil) as the untreated control (0/0). Four biorational insecticides: Aza-direct^®^ (azadirachtin, IRAC group UN, Gowan Company, Yuma, AZ, USA, 1168.60 mL/hectare), Entrust^®^SC (Spinosad, IRAC group 5, Dow AgroSciences LLC, Indianapolis, IN, USA, 584.37 mL/hectare), Grandevo^®^ (*Chromobacterium subtsugae* strain PRAA4-1^T^, IRAC group UN, Marrone Bio Innovations, Davis, CA, USA, 2240.73 gm/hectare), and Spear^®^T (GS-omega/kappa-Hxtx-Hv1a, IRAC group 32, Vestaron, Kalamazoo, MI, USA, 1168.60 mL/hectare) were used in this study to observe their effectiveness in combination with plastic mulch in managing thrips and TCSV. There was an untreated control where we did not use any insecticide. All insecticides were applied as foliar applications once a week, starting from the 3rd week of transplanting and continuing to the 9th week of transplanting. Insecticide treatments were applied weekly using a backpack sprayer (Birchmeier 4 Gallon Backpack Sprayer, model IRIS, Stetten, Switzerland), delivering a volume of 50 to 70 GPA (467.7 to 654.8 L/Hectare), depending on the tomato foliage canopy. The pressure of the sprayer used for spraying was set at 206.84 KPI (2.1 kg/cm^2^). The sprayer had a 50 cm curved brass spray lance with a brass mist nozzle of 1.5 mm to dispense spray materials. The plastic mulch of different types and biorational insecticides tested in this study was arranged in a split-plot design with four replications. Plastic mulches were the main plots, and biorational insecticides were the subplots. The subplots were 4.57 m long and 1.82 m wide. There was a 1.52 m unplanted buffer between the subplots. Thus, each main plot (28.95 m) consisted of five subplots.

### 2.3. Evaluation of Biorational Insecticides and Plastic Mulch Treatments

#### 2.3.1. Sample Collection and Processing for Thrips Separation

The effectiveness of the treatments was evaluated by recording thrips population, TCSV-infected plants, marketable yields, and marketable fruit numbers. To assess thrips management, leaf and flower samples from the treated plots were collected 48 h after each chemical application for up to 10 weeks. Five green, full-grown, and widely open leaves from the top stratum were collected from randomly selected five plants (one leaf/plant) from each plot. We also collected ten widely open flowers from five randomly selected plants (two flowers/plant) from each experimental plot after each chemical application. Flower samples were collected 6 weeks after transplanting and continued up to 10 weeks after transplanting (5 sampling dates). The leaf and flower samples were placed separately into a pint plastic cup (Uline Crystal Clear Plastic Cups-16 oz, Uline 12,575 Uline Drive, Pleasant Prairie, WI 53158) with a thrips-proof lid and marked with the field, row, block, plastic mulch, and plot numbers along with the sampling date. The samples were brought to the Vegetable Entomology Laboratory at TREC and soaked in 70% ethyl alcohol for 20 min to dislodge thrips. The leaves and flowers were then carefully removed from the alcohol, leaving thrips as residue in each cup. The alcohol residue was passed through a sieve (USA Standard Testing Sieve, No. 60, opening 250 micro-meters, Fisher Scientific Company, Waltham, MA, USA) to separate thrips from the alcohol. Thrips collected on the sieve were transferred to a Petri dish (10 cm diam) using a gentle jet of alcohol from a squirt bottle [46]. The number of thrips of different species and their larvae in alcohol was counted using a stereo microscope (10–30×) (Leica Wild M3Z, Micro Optics of Florida, Inc., Plantation, FL, USA). Adult thrips specimens were slide mounted and identified under a digital microscope (VHX-6000; Keyence, Itasca, IL, USA) at 50–200× magnification. Identification of each thrips species was obtained by observing the taxonomic characters, including antennal segments, the position of post-ocellar setae in the ocellar triangle, and the microtrichial comb on the eighth abdominal segment [47,48]. We did not identify the thrips larvae up to their species level.

#### 2.3.2. Marketable Yield

We randomly selected four tomato plants in each plot for harvesting marketable fruits following the US market standard [49]. The marketable fruits (green stage) were collected at 12 weeks after transplanting tomatoes to the field, weighed, and recorded in kilograms for all treated and untreated (control) plots using 31.75 kilograms (70 lb.) capacity scale (CCI Scale Company, Ventura, CA, USA). The weighted fruits were counted separately for each treatment. Tomato plants were carefully inspected for TCSV symptoms and recorded during the time of sample collection each week. We determined the incidence of TCSV based on the symptoms [2]. Infected tomato plants also showed characteristic necrotic ring spots on the fruits. Primarily, the infected leaves were confirmed for TCSV using ImmunoStrip^®^ for tomato spotted wilt virus (TSWV) (Agdia^®^, Inc., Elkhart, IN, USA). Later, the symptomatic leaves were confirmed for TCSV through RT-PCR analysis following the protocol mentioned [50].

### 2.4. Statistical Analysis

The mean number of thrips from each treatment was compared separately for each year. All responses were analyzed using a linear mixed model to account for experimental design (randomized complete block split plot). The random effects were block and block by mulch (PROC GLIMMIX model, SAS Institute, 2013, SAS/STAT 9.3) [51]. Responses that were measured on all sampling dates were averaged over the dates, thus removing any date effect and the large number of zero counts. The resulting means were square root transformed before the analysis was performed. Marketable yield and number of TCSV-infected tomatoes were only measured once per year and, therefore, were not average. Marketable yield still required a squared root transformation, while TCSV counts did not. Transformations were performed to meet the model assumptions. Non-transformed means are reported in the tables. Tukey’s multiple comparisons procedure was used for all post hoc mean comparisons. Degrees of freedom were estimated using the Kenward–Roger’s method. When the F-value for the overall treatment effect was significant, differences of means among treatments (least square means) were separated using Tukey’s multiple comparisons procedure. All the data were analyzed at the 5% level of significance. The Pearson correlation coefficient analysis was conducted to observe the correlation between the variables [52]. 

## 3. Results

### 3.1. Correlation of the Abundance of Thrips, Marketable Yield, Number of Marketable Fruits, and Incidence of TCSV in Tomatoes

Adult thrips and larval thrips populations were negatively correlated to the marketable fruit weight, indicating that an increase in thrips adults and the larval population reduced the number of tomato fruits and marketable yield (Table 1). The correlation between TCSV and marketable fruit weight and TCSV and the number of marketable fruits were also negative. The number of TCSV-infected plants increased with the increase of thrips adults and larvae, indicating a positive correlation. 

### 3.2. The Abundance of Thrips in Tomatoes Based on Leaf Sample

WFT, CBT, and melon thrips (*Thrips palmi* Karny) were commonly found in tomatoes in 2018, with their abundance of 0–10%, 0–24%, and 64–97%, respectively. The data in the tables show the combined number of adults of these three thrips species. In 2018, the adult thrips population was significantly (*F*_4,60_ = 58.04, *p* < 0.0001) lower (0.66 ± 0.08 adult thrips/five leaves) in the Entrust^®^SC treated tomatoes than in other insecticide treatments and untreated control (Table 2). The population of larval thrips was also significantly (*F*_4,60_ = 56.28) lower (0.76 ± 0.07) in Entrust^®^SC treated tomatoes than in other treated and untreated tomato leaves. In 2019, the adult thrips population was comprised of WFT (0–8%), CBT (3–36%), and melon thrips (70–97%). There was no statistical difference in the abundance of adult thrips among the untreated control tomatoes. However, the larval thrips population was significantly (*F*_4,60_ = 9.19, *p* < 0.0001) lower in Entrust^®^SC treated tomatoes than in Spear^®^T and Grandevo^®^ treated and untreated tomatoes. 

In 2018, a significantly lower number (for S/B, 0.45 ± 0.06 adult thrips/five leaves, and for S/W, 0.62 ± 0.08 adult thrips/five leaves) of adult thrips and larval thrips were recorded from treated tomato leaves on S/W and S/B reflective plastic mulches than other plastic mulches and no mulch. In 2019, a significantly (*F*_4,15_ = 6.41, *p* = 0.0032) lower number of adult thrips was found in tomatoes planted on S/B and S/W reflective plastic mulches than on W/B plastic mulch and no mulch. The population of larval thrips was significantly lower in S/B reflective mulch than in W/B mulch, but both treatments along with other treatments, did not differ statistically from the control (Table 2). However, the adult and larval thrips population was impacted by insecticides and mulches, there was no statistical interaction observed between the insecticide and mulch on the population of adult thrips (in 2018, *F*_16,60_ = 1.67, *p* = 0.0794; in 2019, (*F*_16,60_ = 1.47, *p* = 0.1437) and larval thrips (in 2018, *F*_16,60_ = 1.06, *p* = 0.4089; in 2019, *F*_16,60_ = 1.12, *p* = 0.3543).

### 3.3. The Abundance of Thrips in Tomatoes Based on Flower Sample

Adult thrips population collected from flower samples of tomatoes consisted of WFT (0–18%), CBT (12–37%), and melon thrips (51–82%) in 2018. Entrust^®^SC treated tomatoes had a significantly lower number of adult thrips than the other insecticide treatments and the untreated control in 2018 (*F*_4,72_ = 82.00, *p* < 0.0001) (Table 3). A significantly (*F*_4,60_ = 5.83, *p* = 0.0005) lower number of larval thrips was observed in tomatoes treated with Spear^®^T, Aza-direct^®^, and Entrust^®^SC than in the untreated control. In 2019, the mean number of adult thrips on Entrust^®^SC treated tomatoes was significantly (*F*_4,60_ = 8.91, *p* < 0.0001) lower than the untreated control. However, the population of larval thrips did not significantly differ between the insecticide treatments and the untreated control in 2019.

Adult and larval thrips populations were also significantly (*F*_4,12_ = 161.18, *p* < 0.0001 for adults and *F*_4,12_ = 33.08, *p* < 0.0001 for larva) lower in tomatoes treated with reflective plastic mulches (S/B and S/W) and other plastic mulches (W/B and B/B) than with no mulch in 2018. There was no statistical difference between the mulches and no mulch in the abundance of thrips in tomato flowers in 2019 (Table 3). 

There was no statistical interaction between the mulch and insecticide on the abundance of adult thrips in flowers in both years (in 2018, *F*_16,60_ = 1.22, *p* = 0.2733; in 2019, *F*_16,60_ = 1.11, *p* = 0.3697). The interaction of mulch and insecticide was significant (*F*_16,60_ = 2.63, *p* = 0.0036) on the abundance of larval thrips in flowers in 2018 but not in 2019 (*F*_16,60_ = 0.71, *p* = 0.7734).

### 3.4. Marketable Yield

In 2018, Entrust^®^SC treated tomatoes produced a significantly (*F*_4,60_ = 42.94, *p* < 0.0001) higher marketable fruit weight than other insecticide-treated and untreated tomatoes (Table 4). The number of fruits in the treated and untreated tomatoes followed a similar pattern to the marketable fruit weight (*F*_4,60_ = 36.20, *p* < 0.0001). Tomatoes treated with Entrust^®^SC produced a significantly (for marketable fruit weight, *F*_4,60_ = 6.42, *p* = 0.0002, and for number of fruits, *F*_4,60_ = 5.40, *p* = 0.0009) higher marketable fruit weight and the number of fruits than the untreated control in 2019 (Table 4).

Tomatoes grown on the S/B and S/W reflective plastic mulches and B/B plastic mulch produced significantly (*F*_4,15_ = 12.89, *p* < 0.0001) higher marketable fruit weight than other mulch and no mulch in 2018. A significantly (*F*_4,15_ = 11.73, *p* = 0.0002) higher number of fruits was recorded from tomatoes treated with the S/B reflective mulch than the untreated control. In 2019, tomatoes treated with the S/B plastic mulch produced a significantly (*F*_4,15_ = 3.32, *p* = 0.0390) higher marketable fruit weight than in no mulch. The number of fruits in the same year followed the same trend as the marketable fruit weight (*F*_4,15_ = 3.41, *p* = 0.0358). The marketable yield as marketable fruit weight and the number of marketable fruits did not show any interaction between the mulch and insecticide (for marketable fruit weight in 2018, *F*_16,60_ = 1.55, *p* = 0.1136 and in 2019, *F*_16,60_ = 0.98, *p* = 0.4868; for number of marketable fruit in 2018, *F*_16,60_ = 1.45, *p* = 0.1507; in 2019, (*F*_16,60_ = 1.07, *p* = 0.4046)).

### 3.5. Incidence of TCSV

In 2018, a significantly lower number of TCSV-infected tomatoes was observed in Entrust^®^SC, Spear^®^ T, and Aza-direct^®^ treated plants than in the untreated control plants (*F*_4,60_ = 26.65, *p* < 0.0001). However, the incidence of TCSV did not show any significant difference between the insecticide treatments and untreated control in 2019 (Figure 1).

A significantly lower number of TCSV-infected tomatoes was recorded from S/B and S/W reflective plastic mulches and B/B plastic mulch than no mulch in 2018 (*F*_4,12_ = 9.2129, *p* = 0.0012). However, in 2019, reflective plastic mulch and other plastic mulch-treated tomatoes did not differ statistically from the no mulch on the incidence of TCSV (Figure 2). There was no statistical interaction between the mulch and insecticide on the incidence of TCSV-infected plants in both years (in 2018, *F*_16,60_ = 1.26, *p* = 0.2514; in 2019, *F*_16,60_ = 1.71, *p* = 0.0683).

## 4. Discussion

Thrips can transmit tospovirus within a short feeding period [53]. The use of conventional insecticides cannot provide the ultimate control over tospovirus when it is transmitted by migratory vectors coming from the outside of the field [28,54,55]. Moreover, the rapid development of resistance to insecticides reduces the conventional insecticides’ efficacy as foliar applications [56]. Repellent strategies using reflective mulch, at least at the border rows of crop fields, would be the most effective to save the crop from primary infection of tospovirus [31,57]. At the same time, insecticides with the ability to control larval populations would likely be the most effective in reducing secondary infection from TCSV [41]. In our field study, we used four biorational insecticides with repellent and larval control activity to manage thrips. The liquid concentrate of Grandevo^®^ was used, which functions primarily as a central nervous system inhibitor of the target pest (Specimen label, Vestaron Corporation, Durham, NC, USA) [58]. The mode of action of Aza-direct^®^ is contact or ingestion, working as a repellent, antifeedant, and interfering with the molting process (Specimen label, Gowan company, Yuma, AZ, USA) [59]. The fermentation solids of Spear^®^T function as a stomach poison, feeding deterrent, and reduce fecundity and oviposition (Specimen label, Marrone Bio Innovations, Raleigh, NC, USA). Entrust^®^ SC is a biologically derived product from the fermentation of *Saccharopolyspora spinosa* (Specimen label, Dow AgroSciences, Midland, MI, USA), altering the function of nicotinic and GABA-gated ion channels, causing rapid excitation of the insect nervous system, leading to involuntary muscle contractions, tremors, paralysis, and death [60].

Entrust^®^ SC was found to be useful among the bio-rational insecticides in the present study to manage both adult and larval thrips, as revealed from the tomato leaf sampling in 2018 and 2019. The same insecticide was found to be effective in reducing the adult thrips and larval thrips abundance in tomato flowers in 2018. There was little or no impact of the insecticidal treatments on the abundance of adult or larval thrips in tomato flowers in 2019. The overall trend was lower numbers of adult and larval thrips in tomato flowers from Entrust^®^ SC treatment compared to other biorational insecticides and untreated control in 2019. Entrust^®^ SC has been documented to be the most potent biorational insecticide for the management of western flower thrips in other studies [61,62,63]. Moreover, spinosad is suitable to use in rotation with other insecticides [64] and in combination with the natural enemies of the thrips [65]. The Spear^®^T and Grandevo^®^ treatments in our study did not show satisfactory results in managing the thrips population in tomato, which is consistent with the other research findings [66,67]. In the present study, we revealed a low population of adult thrips and larvae on the silver reflective plastic mulches compared to the other plastic mulches and no mulch, irrespective of tomato leaves and flowers in both 2018 and 2019. Several research studies indicated the advantages of reflective mulches to reducing thrips population in host plants, including row vegetable crops such as tomato and capsicum [31,32,36,38,68,69,70]. The use of plastic mulch for row crops improved marketable yield, suppressed weeds, and increased the efficient use of water and fertilizers [71].

In the current study, Entrust^®^ SC treated tomatoes produced a higher marketable yield and higher number of marketable fruits compared to other insecticides. Perhaps the effectiveness of Entrust^®^SC against the lepidopteran pests might have added to the marketable weight. Tomato plants on both reflective plastic mulches (‘S/B’ and ‘S/W’) produced significantly higher yields and a higher number of marketable fruits than other mulches and no mulch. Both plastic mulch and biorational insecticides significantly increased the marketable yield of tomatoes in 2018. Our study coincided with another researcher who reported that the marketable yield of tomatoes could be increased by using UV reflective much, eliminating the need for complete reliance on insecticides and preventing the development of resistance in thrips vectors [72]. Our study also documents that the silver plastic mulches were shown to significantly reduce the number of virus-infected tomatoes compared to no mulch. Our results were consistent with the previous studies indicating that the tomato plants grown on silver mulches had a lower incidence of TCSV [40] and TSWV (related to TCSV) than plants grown on non-UV reflective plastic mulches [38,39]. It can be argued that silver mulches modify the light environment around the plants without causing any phytotoxicity and thus reduce the thrips activity [28,57,73].

Early season infection of tospovirus in our research field was observed probably because of the invasion of migratory thrips from nearby infected weeds or wild plants [4,5,36]. Plants infected with tospovirus evidenced stunting, leaf necrosis, and death of terminal shoot and stem, retarding the growth of the plant and reducing the marketable yield [12,28,74]. We found that the primary infection of tospovirus can be restrained by using silver plastic mulch or UV-reflective plastic mulch, which can lead to control over the secondary infection. However, the reflective mulch’s efficacy diminished as the growing plants began to cover the bed surface, reflecting less light from the mulch and repelling fewer thrips [65,72,75]. Thus, insecticides, as one IPM tactic, could be an important tool to inhibit the secondary spread of diseases within a tomato field [32,76,77].

## 5. Conclusions

TSW-resistant cultivars based on a single SW5 gene offer promise for TSWV management [78,79,80]. The same cultivars have the potential to minimize the yield loss by TCSV since both TCSV and TSWV are classified under the same virus group (Family: Bunyaviridae, genus *Tospovirus*) [2,12]. However, dependency on a single gene for resistance can produce high selection pressure on local TSWV or TCSV isolates, leading to resistance failure [81,82,83]. Insecticides and reflective mulches can be the other prospective management tactics as part of integrated management programs for TCSV [54,84,85,86]. Our study provided further evidence that the use of cultural (reflective plastic mulch) and chemical (biorational insecticides) tools were effective in significantly reducing the thrips population and TCSV-infected tomatoes, thus increasing the marketable yield (number and weight) compared with standard tomato production practices.

## Figures and Tables

**Figure 1 insects-14-00740-f001:**
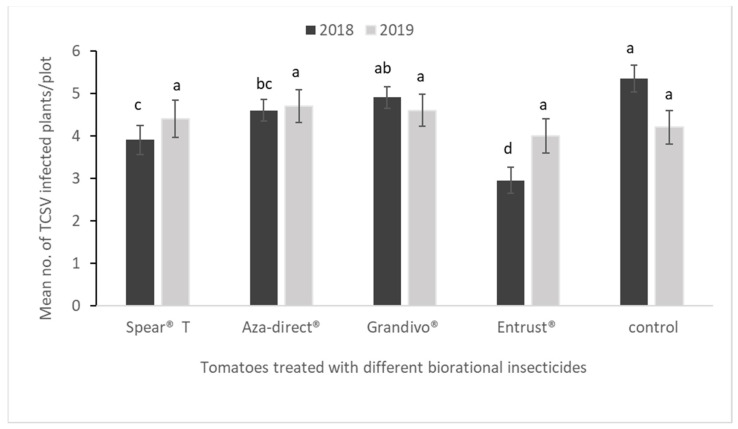
Tomato chlorotic spot virus-infected tomato plants per plot (*F*_4,60_ = 26.65, *p* < 0.0001, 2018 and *F*_4,60_ = 0.9752, *p* < 0.4279, 2019) treated with different biorational insecticides in two years. Data presented are mean ± standard error (SE).

**Figure 2 insects-14-00740-f002:**
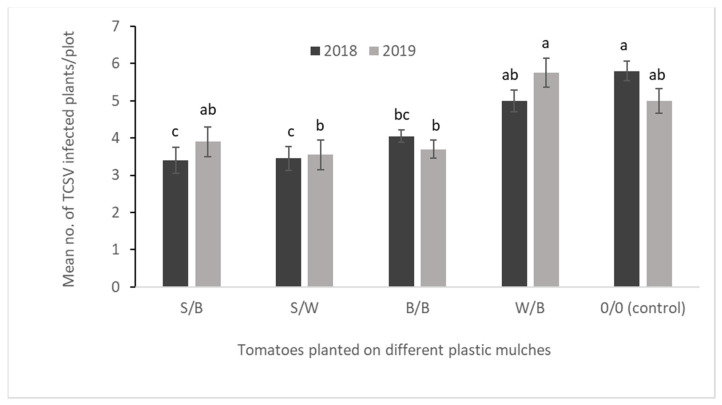
Tomato chlorotic spot virus-infected tomato plants per plot (*F*_4,12_ = 9.2129, *p* = 0.0012, 2018 and *F*_4,12_ = 4.4481, *p* = 0.0196, 2019) treated with different plastic mulches for two years. Data presented are mean ± standard error (SE). S/B = silver on black mulch, S/W = silver on white mulch, B/B= black on black mulch, W/B = white on black, 0/0 = no mulch (control).

**Table 1 insects-14-00740-t001:** Pearson correlation (r values) and *p*-values for various parameters (total adult thrips in leaves, thrips larvae in leaves, total adult thrips in flowers, thrips larvae in flowers, marketable yield, number of marketable fruits and number of TCSV-infected plants in tomato). (Range of N values 120).

Variables	Adult Thrips in Leaves	Thrips Larvae in Leaves	Adult Thrips in Flowers	Thrips Larvae in Flowers	TCSV	Marketable Yield	No of Marketable Fruits	
Adult thrips in leaves		0.7657, <0.0001	0.8791, <0.0001	0.6682, <0.0001	0.3851, <0.0001	−0.2689, 0.0002	−0.2753, 0.0002	
Thrips larvae in leaves	0.7657, <0.0001		0.7802, <0.0001	0.6847, <0.0001	0.4252, <0.0001	−0.3840, <0.0001	−0.3959, 0.0001	
Adult thrips in flowers	0.8791, <0.0001	0.7802, <0.0001		0.7284, <0.0001	0.3358, <0.0001	−0.3079, <0.0001	−0.3012, <0.0001	
Thrips larvae in flowers	0.6682, <0.0001	0.6847, <0.0001	0.7284, <0.0001		0.2977, <0.0001	0.2406, 0.0006	−0.2677, 0.0001	
TCSV	0.3851, <0.0001	0.4252, <0.0001	0.3358, <0.0001	0.2977, <0.0001		−0.4836, <0.0001	−0.4532, <0.0001	
Marketable yield	−0.2589, 0.0002	−0.3840, <0.0001	−0.3079, <0.0001	−0.2406, 0.0006	−0.4836, <0.0001		0.9571, <0.0001	
No. of marketable fruits	−0.2753, 0.0002	−0.3959, 0.0001	−0.3012, <0.0001	−0.2677, 0.0001	−0.4532, <0.0001	0.9571, <0.0001		

**Table 2 insects-14-00740-t002:** Mean ± standard error (SE) number of adult thrips per five tomato leaves collected from each plot treated with different plastic mulches and biorational insecticides.

Treatments	Mean ± Standard Error (SE) Number of Thrips per Five Tomato Leaves			
	2018		2019	
Insecticides	Adult	Larva	Adult	Larva
Spear^®^ T	2.17 ± 0.21ab ^z^	1.75 ± 0.15b	0.66 ± 0.09	0.87 ± 0.15ab
Aza-direct^®^	1.77 ± 0.18b	1.93 ± 0.21b	0.45 ± 0.05	0.56 ± 0.09bc
Grandevo^®^	2.15 ± 0.21ab	2.18 ± 0.15b	0.06 ± 0.08	0.99 ± 0.20ab
Entrust^®^SC	0.66 ± 0.08c	0.76 ± 0.07c	0.50 ± 0.08	0.26 ± 0.04c
Untreated control	2.39 ± 0.22a	3.40 ± 0.19a	0.58 ± 0.08	1.12 ± 0.17a
Statistics	*F*_4,60_ = 58.04	*F*_4,60_ = 56.28	*F*_4,60_ = 1.95	*F*_4,60_ = 9.19
	*p* < 0.0001	*p* < 0.0001	*p* = 0.1140	*p* < 0.0001
Mulches				
S/B	0.45 ± 0.06c	1.15 ± 0.21c	0.29 ± 0.05c	0.42 ± 0.08b
S/W	0.62 ± 0.08c	1.14 ± 0.11c	0.28 ± 0.04c	0.65 ± 0.11ab
B/B	2.39 ± 0.20b	2.02 ± 0.12b	0.49 ± 0.06abc	0.66 ± 0.10ab
W/B	2.43 ± 0.20ab	2.42 ± 0.15b	0.96 ± 0.11a	1.11 ± 0.22a
0/0	3.25 ± 0.25a	3.30 ± 0.18a	0.78 ± 0.09ab	0.95 ± 0.16ab
Statistics	*F*_4,12_ = 83.88	*F*_4,12_ = 43.86	*F*_4,15_ = 6.41	*F*_4,15_ = 3.31
	*p* < 0.0001	*p* < 0.0001	*p* = 0.0032	*p* = 0.0394

^z^ Means within the same column followed by the same letter are not significantly different at *p* ≤ 0.05, according to Tukey’s HSD test.

**Table 3 insects-14-00740-t003:** Mean ± standard error (SE) number of adult thrips per ten tomato flowers collected from each plot treated with different plastic mulches and biorational insecticides.

Treatments	Mean ± Standard Error (SE) Number of Thrips per Ten Tomato Flowers			
	2018		2019	
Insecticides	Adult	Larva	Adult	Larva
Spear^®^ T	4.10 ± 0.23b ^z^	0.83 ± 0.10b	1.87 ± 0.16ab	0.27 ± 0.06a
Aza-direct^®^	4.49 ± 0.28b	0.79 ± 0.11b	2.18 ± 0.22a	0.20 ± 0.06a
Grandevo^®^	4.84 ± 0.30b	1.08 ± 0.14ab	2.49 ± 0.25a	0.37 ± 0.17a
Entrust^®^SC	1.96 ± 0.15c	0.80 ± 0.11b	1.36 ± 0.14b	0.05 ± 0.02a
Untreated control	5.98 ± 0.31a	1.36 ± 0.15a	2.40 ± 0.20a	0.15 ± 0.04a
Statistics	*F*_4,72_ = 82.00	*F*_4,60_ = 5.83	*F*_4,60_ = 8.91	*F*_4,72_ = 2.57
	*p* < 0.0001	*p* = 0.0005	*p* < 0.0001	*p* = 0.0448
Mulches		2018		2019
S/B	2.07 ± 0.15c	0.45 ± 0.07cd	1.69 ± 0.17a	0.20 ± 0.06a
S/W	2.11 ± 0.17c	0.35 ± 0.06d	1.65 ± 0.21a	0.16 ± 0.05a
B/B	5.03 ± 0.28b	0.82 ± 0.11bc	2.52 ± 0.22a	0.18 ± 0.05a
W/B	5.12 ± 0.24b	1.22 ± 0.13b	2.23 ± 0.22a	0.15 ± 0.04a
0/0	7.05 ± 0.27a	2.01 ± 0.16a	2.22 ± 0.17a	0.35 ± 0.16a
Statistics	*F*_4,72_ = 161.18	*F*_4,12_ = 33.08	*F*_4,12_ = 3.25	*F*_4,72_ = 0.37
	*p* < 0.0001	*p* < 0.0001	*p* = 0.0503	*p* = 0.8263

^z^ Means within the same column followed by the same letter are not significantly different at *p* ≤ 0.05, according to Tukey’s HSD test.

**Table 4 insects-14-00740-t004:** Mean ± standard error (SE) number of marketable fruit weight in kg and number of fruits per four tomato plants from each plot treated with different plastic mulches and biorational insecticides.

Treatments	Mean ± Standard Error (SE) Number of Marketable Fruit Weight in kg and Fruits/Four Tomato Plants			
	2018		2019	
Insecticides	Marketable fruit weight	No. of fruit	Marketable fruit weight	No. of fruit
Spear^®^ T	4.18 ± 0.48b ^z^	18.90 ± 2.06b	3.42 ± 0.61abc	15.05 ± 2.40ab
Aza-direct^®^	4.26 ± 0.52b	19.65 ± 2.32b	4.67 ± 0.75ab	22.00 ± 3.17a
Grandevo^®^	2.94 ± 0.38c	13.30 ± 1.82c	2.87 ± 0.59bc	12.90 ± 2.25b
Entrust^®^SC	7.27 ± 0.70a	33.35 ± 3.34a	5.57 ± 0.72a	24.30 ± 3.38a
Untreated control	2.16 ± 0.28c	10.70 ± 0.48c	2.44 ± 0.47c	12.70 ± 2.34b
Statistics	*F*_4,60_ = 42.94	*F*_4,60_ = 36.20	*F*_4,60_ = 6.42	*F*_4,60_ = 5.40
	*p* < 0.0001	*p* < 0.0001	*p* = 0.0002	*p* = 0.0009
Mulch				
S/B	6.22 ± 0.79a	28.45 ± 3.49a	5.84 ± 0.95a	25.35 ± 3.98a
S/W	5.12 ± 0.59ab	23.50 ± 2.93ab	3.95 ± 0.64ab	18.65 ± 3.10ab
B/B	4.25 ± 0.49ab	19.90 ± 2.21ab	4.25 ± 0.57ab	19.35 ± 2.31ab
W/B	3.28 ± 0.38bc	15.10 ± 1.89bc	2.56 ± 0.40ab	12.00 ± 1.68ab
0/0	1.94 ± 0.30c	8.95 ± 1.30c	2.38 ± 0.41b	11.60 ± 1.84b
Statistics	*F*_4,15_ = 12.89	*F*_4,15_ = 11.73	*F*_4,15_ = 3.32	*F*_4,15_ = 3.41
	*p* < 0.0001	*p* = 0.0002	*p* = 0.0390	*p* = 0.0358

^z^ Means within the same column followed by the same letter are not significantly different at *p* ≤ 0.05, according to Tukey’s HSD test.

## Data Availability

The data presented in this study are available on request from the corresponding author. The data is not publicly available due to the funding organization’s choice.

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
