# Peer review of "Integrated Effect of Plastic Mulches and Biorational Insecticides in Managing Tomato Chlorotic Spot Virus (TCSV) and Its Vector Thrips in Tomatoes"

_insects, 2023, doi:10.3390/insects14090740_

Round 1

Reviewer 1 Report

The issue raised by the authors is indeed an important one, and is interesting from both scientific and practical perspectives. The topic may be of interest to readers of journal. The structure of the article does meet the requirements of the publication (Research Manuscript Sections).

The introduction does contain a clear statement of the problem, the relevant literature on the subject, and a proposed approach or solution. The topicality and novelty of the research are understood from the introduction. The literature review refers to the central issue of the paper, it is quite extensive, relevant and thorough. The review will be of interest to other researchers. I would like to mention that the authors have comprehensively studied the literature on the issue. References are correct.

The manuscript is written with detailed methodology and results.

The conclusion is consistent with presented arguments and evidence. The results complete previous results on the matter and are supported by references.

I would recommend that the authors reduce the annotation to 200 words indicate the period of analysis in the abstract, as well as list which research methods were used.

Although the authors have studied the literature on this problem quite well, nevertheless, there are only two sources published in the work over the past five years. That's not enough.  Authors should study the literature on the topic published over the past five years even more thoroughly.

The article may be of interest to readers of the journal.

Author Response

Point 1. I would recommend that the authors reduce the annotation to 200 words indicate the period of analysis in the abstract, as well as list which research methods were used.

Response 1: The abstract is changed accordingly,

In the USA, tomato chlorotic spot virus (TCSV) was first identified in Miami-Dade County of Florida in 2012. This viral disease is transmitted by thrips (Thysanoptera: Thripidae) of different species, imposing a serious threat to the entire tomato production in the state. Both cultural and chemical control techniques could be essential tools to combat this vector-borne disease. In the present two year long study, we determined the effect of different types of plastic mulches, and biorational insecticides, on managing thrips and TCSV. Results from the leaf  and flower samples showed a significantly lower adult thrips population in Entrust®SC treated tomatoes than other treated and untreated tomatoes in 2018. Silver on black and silver on white reflective plastic mulches significantly reduced the adult thrips population in 2018.s . In both study years, marketable yield was significantly higher in tomatoes treated with Entrust®SC and reflective plastic mulches than other treatments. The incidence of TCSV was significantly reduced in tomatoes treated with Entrust®SC and reflective plastic mulchesthan the untreated control in 2018. The interaction between mulch and insecticide on the incidence of TCSV infected plants in both years was significant. Marketable yield was negatively correlated with the thrips population as observed from the Pearson correlation coefficient analysis. This research describes a potentially viable management program for thrips and thrips transmitted TCSV.

Point 2. Although the authors have studied the literature on this problem quite well, nevertheless, there are only two sources published in the work over the past five years. That's not enough.  Authors should study the literature on the topic published over the past five years even more thoroughly.

Response 2: Added some references (Ref. No. 37, 42)

Reviewer 2 Report

The main criticism is that the statistics on interactions for the split plot design were not included in the tables and were inconsistently referred to in the text. Also, it appears the authors were not consistent in what was referred to as a significant (e.g., Table 3 P=0.0448 or even P=0.0503 not significant or did you mean that the Tukey test did not separate out the means?). Also, I do not prefer to have statistical tests reported in the abstract (e.g., lines 27, 31, 34), that should be in the results section only. Other suggested changes are as follows.

Line 15, change to "can be a potential management approach for this pest."

Line 37-38, change to "This research describes a potentially viable management program for thrips and ..."

Lines 214 and 226, did you mean over both years, because the table appears to be both years and you need to add the N value (or range of N values) in the footnote to Table 1.

Lines 249-251 and lines 314-316, avoid single sentence paragraphs, just add to the previous paragraph.

Line 251 where are the interaction statistics? This should be included in Table 2 as statistics for a Mainplot x Subplot interaction.

Line 266, are you referring to Table 3 and where are the Tukey mean separation values, if not significant then all means should be followed by an "a". In Table 3, the Larva 2019 column appears to be significant P<0.05, but no values are given. You do state 5% level of significance in the M&M section.

Line 282, don't repeat the 7.2... mean, just cite the Table. Also, why not talk about % differences, especially when it come to crop yield so a reader can translate the results to potential yield gains with your best treatment program.

Line 307, insert "and SpaerT" before treated plants.

Lines 308 and 311, insert the year after "However", e.g. "However in 2019,"

Line 332, change to "would likely be the most effective to reduce primary..."

Line 334, replace "are" with "would likely be"

Line 352, replace "Though, numerically..." with "The overall trend was lower numbers of adult and larval thrips in tomato flowers..."

Line 367, did Entrust control other pests like Lepidoteran larvae that might have added to marketable weight?

Line 382, replace "crop" with "our research" and insert "probably" after "observed".

Line 386, begin the sentence with "We found that..."

Lines 390-391, replace with "Thus, insecticides as one IPM tactic, could be an important tool to inhibit the secondary spread..."

Line 590, it's "Riley, D.G." !

Author Response

Comments and Suggestions for Authors

Point 1. The main criticism is that the statistics on interactions for the split plot design were not included in the tables and were inconsistently referred to in the text.

Response 1: The interactions of main plot and subplot was described accordingly after each result section. The results did not presented in a tabular form just to avoid the clumsiness in the manuscript.

Point 2. Also, it appears the authors were not consistent in what was referred to as a significant (e.g., Table 3 P=0.0448 or even P=0.0503 not significant or did you mean that the Tukey test did not separate out the means?).

Response 1: The letters are added in table 3.

 Point 3. Also, I do not prefer to have statistical tests reported in the abstract (e.g., lines 27, 31, 34), that should be in the results section only.

Response 1: The statistical tests are deleted from the abstract.

Other suggested changes are as follows.

Point 4. Line 15, change to "can be a potential management approach for this pest."

Response 1: Changed accordingly.

“A combined effect of cultural and chemical control strategy can be potential management ap-proach for implemented to manage this pest without imposing any environmental stress”

Point 5. Line 37-38, change to "This research describes a potentially viable management program for thrips and ..."

Response 1: Changed according to the reviwer’s suggestions

“This research describes a potentially viable management program information can help growers to develop sustainable management programs for thrips and thrips transmitted TCSV”.

Point 6. Lines 214 and 226, did you mean over both years, because the table appears to be both years and you need to add the N value (or range of N values) in the footnote to Table 1.

Response 1: N values were 120. Added.

Point 7. Lines 249-251 and lines 314-316, avoid single sentence paragraphs, just add to the previous paragraph.

Response 7: Combined with the previous paragraph.

Point 8. Line 251 where are the interaction statistics? This should be included in Table 2 as statistics for a Mainplot x Subplot interaction.

Response 8: The interaction statistics are described in the text to avoid to many tables in the manuscript.

Point 9. Line 266, are you referring to Table 3 and where are the Tukey mean separation values, if not significant then all means should be followed by an "a". In Table 3, the Larva 2019 column appears to be significant P<0.05, but no values are given. You do state 5% level of significance in the M&M section.

Response 9: Added letters.

Point 10. Line 282, don't repeat the 7.2... mean, just cite the Table. Also, why not talk about % differences, especially when it come to crop yield so a reader can translate the results to potential yield gains with your best treatment program.

Response 10: Deleted the number values.

Point 11. Line 307, insert "and SpaerT" before treated plants.

Response 11: Added.

“In 2018, a significantly lower number of TCSV-infected tomatoes was observed in En-trust®SC and Spear® T treated plants than the untreated control plants”

Point 12. Lines 308 and 311, insert the year after "However", e.g. "However in 2019,"

Response 12: Changed.

 “However, in 2019, reflective mulch treated tomatoes did not differ statistically from the no mulch in 2019 on the incidence of TCSV”

Point 13. Line 332, change to "would likely be the most effective to reduce primary..."

Response 13: Changed like following.

“At the same time, insecticides with the ability to control larval populations would likely beare the most feasible effective to prevent reduce secondary infection from TCSV”

Point 14. Line 334, replace "are" with "would likely be"

Response 14: Replaced as follows.

“At the same time, insecticides with the ability to control larval populations would likely beare the most feasible effective to prevent reduce secondary infection from TCSV”

Point 15. Line 352, replace "Though, numerically..." with "The overall trend was lower numbers of adult and larval thrips in tomato flowers..."

Response 15: Replaced accordingly.

“The overall trend was lower numbers of adult and larval thrips in tomato flowers from Entrust® SC treatment compared to other biorational insecticides and untreated control in 2019”

Point 16. Line 367, did Entrust control other pests like Lepidoteran larvae that might have added to marketable weight?

Response 16: Added the following line.

“Perhaps the effectiveness of Entrust®SC against the lepidoptern pests might have added to marketable weight”

Point 17. Line 382, replace "crop" with "our research" and insert "probably" after "observed".

Response 17: Added in the manuscript.

“Early season infection of tospovirus in crop our research field was observed probably because of the invasion of migratory thrips from nearby infected weeds or wild plants”

Point 18. Line 386, begin the sentence with "We found that..."

Response 18: Added.

Point 19. Lines 390-391, replace with "Thus, insecticides as one IPM tactic, could be an important tool to inhibit the secondary spread..."

Response 19: Replaced the sentence as

“Thus, insecticides as one IPM tactic, could be  important tool to inhibit the secondary spread of diseases within a tomato field [31,73,74].”

Point 20. Line 590, it's "Riley, D.G." !

Response 20: Corrected.

Reviewer 3 Report

The work is well written but some points can be improved. These are:

Line 34: I know that there is no interaction between biocides and mulches  however the best-performing mulches could be given here as well

Line 46: Do you have another literature supporting that tomato plants are infected with the TCSV after 3 weeks of transplatation?

line 93, no direct relationship with the previous paragraph please re-formulate the sentence

Line 96- Please add why it is not possible to combat thrips using mulches in the late stage of the plants. Do you cover just the soil with mulches?

Line 201: What do you mean by block and block my much as random effect? please explain it

Moreover, You did not mention which statistical program/software you used to analyse the data

Table 1: As a reader, it is difficult to understand and interpret the table. It would be great if you add colours like in the Pearson correlation

Line 379: You argue that reflective silver mulches provide control for thrips, indirectly the virus. You tested fruit quality but I am also wondering if this mulch reflects light, does the vegetative organs of the plant turn pale? or if you have observed that the plant needs more irrigation due to using these mulches, could you add these observations? Fascinating results, We only use black plastic as mulch

Author Response

Point 1. Line 34: I know that there is no interaction between biocides and mulches  however the best-performing mulches could be given here as well

Response 1: Added in the abstract.

“The incidence of TCSV was significantly reduced in tomatoes treated with Entrust®SC (F4,60=28.46, P<0.0001) and silver on black reflective plastic mulches (F4,12=7.37, P=0.0031) than the untreated control in 2018”

Point 2. Line 46: Do you have another literature supporting that tomato plants are infected with the TCSV after 3 weeks of transplatation?

Response 2: Added (Ref. No. 3).

Point 3. Line 93, no direct relationship with the previous paragraph please re-formulate the sentence

Response 3: Organize the sentences as follows.

“The use of metalized and ultra-violet (UV) reflective plastic mulches can also be an effective way to manage thrips, tospovirus prevalence, and increase the yield per unit area [30,31,32,33,34,35]. Moreover, the use of plastic mulch to cover the beds is an effective tool for migrating thrips population only at the early developmental stage of plants with small canopy”

Point 4. Line 96- Please add why it is not possible to combat thrips using mulches in the late stage of the plants. Do you cover just the soil with mulches?

Response 4: Explained as

“At the late growth stage of tomato with dense canopy, reflection capacity of plastic mulch becomes greatly limited. Metalized plastic mulch showed a lower incidence of TSWV in tomatoes than the plants grown on black plastic mulch”

Point 5. Line 201: What do you mean by block and block my much as random effect? please explain it

Response 5: According to our statistician, J. Colee, these random effects were included to account for the design of the experiment. It explains how the design was accounted for the model.

Point 6. Moreover, You did not mention which statistical program/software you used to analyse the data

Response 6: Added.

The random effects were block and block by mulch (PROC GLIMMIX model, SAS Institute, 2013, SAS/STAT 9.3)

Point 7. Table 1: As a reader, it is difficult to understand and interpret the table. It would be great if you add colours like in the Pearson correlation

Response 7: Table 1 is explained in the result section.

Point 8. Line 379: You argue that reflective silver mulches provide control for thrips, indirectly the virus. You tested fruit quality but I am also wondering if this mulch reflects light, does the vegetative organs of the plant turn pale? or if you have observed that the plant needs more irrigation due to using these mulches, could you add these observations? Fascinating results, We only use black plastic as mulch

Response 8: Added.

“It can be argued that silver mulches modify the light environment around the plants without causing any phytotoxicity and thus reduce the thrips activity”